# Long-Term Evaluation of Biomarkers in the Czech Cohort of Gaucher Patients

**DOI:** 10.3390/ijms241914440

**Published:** 2023-09-22

**Authors:** Věra Malinová, Helena Poupětová, Martin Řeboun, Lenka Dvořáková, Stella Reichmannová, Ivana Švandová, Lenka Murgašová, David C. Kasper, Martin Magner

**Affiliations:** 1Department of Pediatrics and Inherited Metabolic Disorders, First Faculty of Medicine, Charles University and General University Hospital, 128 08 Prague, Czech Republichpoup@lf1.cuni.cz (H.P.); martin.reboun@vfn.cz (M.Ř.); lenka.dvorakova@vfn.cz (L.D.); stella.reichmannova@vfn.cz (S.R.); ivasvand@gmail.com (I.Š.); lenka.murgas@gmail.com (L.M.); 2ARCHIMEDlife Laboratories, 1110 Vienna, Austria; d.kasper@archimedlife.com

**Keywords:** type 1 Gaucher disease, glucosylsphingosine, lyso-Gb1, chitotriosidase, long term therapy

## Abstract

A personalized treatment decision for Gaucher disease (GD) patients should be based on relevant markers that are specific to GD, play a direct role in GD pathophysiology, exhibit low genetic variation, reflect the therapy, and can be used for all patients. Thirty-four GD patients treated with enzyme replacement therapy (ERT) or substrate reduction therapy (SRT) were analyzed for platelet count, chitotriosidase, and tartrate-resistant acid phosphatase activity in plasma samples, and quantitative measurement of Lyso-Gb1 was performed in dried blood spots. In our ERT and SRT study cohorts, plasma lyso-GL1 correlated significantly with chito-triosidase (ERT: r = 0.55, *p* < 0.001; SRT: r = 0.83, *p* < 0.001) and TRAP (ERT: r = 0.34, *p* < 0.001; SRT: r = 0.88, *p* < 0.001), irrespective of treatment method. A platelet count increase was associated with a Lyso-Gb1 decrease in both treatment groups (ERT: *p* = 0.021; SRT: *p* = 0.028). The association of Lyso-Gb1 with evaluated markers was stronger in the SRT cohort. Our results indicate that ERT and SRT in combination or in a switch manner could offer the potential of individual drug effectiveness for particular GD symptoms. Combination of the key biomarker of GD, Lyso-Gb1, with other biomarkers can offer improved response assessment to long-term therapy.

## 1. Introduction

Gaucher disease (GD, OMIM #230800), one of the most common lysosomal storage disorders, is caused by biallelic pathogenic variants in the *GBA* gene. This autosomal recessive disorder due to an enzymatic deficiency of β-glucocerebrosidase (EC 3.2. 1.45) results in an intra-lysosomal accumulation of glucosylceramide (GlcCer, glucosylcerebroside). Glucosylceramide accumulates primarily within cells of mononuclear phagocyte origin in the spleen, liver, and bone marrow. During its deacylation by acid ceramidase, glucosylceramide transforms into a sensitive biomarker for GD, lyso-glucosylsphingosine (lyso-Gb1). The accumulation of deacylated lysolipids leads to a progressive disease hallmarked by immune dysregulation and multi-system involvement [1,2,3].

The clinical manifestation of GD varies broadly from a perinatal-lethal form to an asymptomatic form. The classification of GD by clinical subtype is useful for prognosis and disease management. Three major clinical types are distinguished by the absence (GD type 1) or presence (GD types 2 and 3) of neurologic signs and symptoms. Perinatal-lethal and cardiovascular forms represent fewer common phenotypes. Symptoms of the most prevalent type 1 GD may include hepatosplenomegaly, anemia, thrombocytopenia, growth delay, and bone or pulmonary involvement. Most heterozygous mutations in the GBA1 gene elevate the risk of Parkinson’s disease and dementia with Lewy bodies. Even though type 1 GD patients classically do not have CNS involvement, they are at increased risk for developing parkinsonism and Parkinson’s disease (PD). Heterozygous mutations in the GBA1 gene cause a more severe PD phenotype and are associated with synucleinopathies in general (Thaler et al., 2017 [4]). However, large-scale studies report that only 8% to 12% of type 1 GD patients show PD symptoms at age 80 years (Blauwendraat et al., 2023 [5]). The rimary neurologic disease of type 2 and 3 GD is characterized by an earlier age of onset and the rate of disease progression, a neurologic involvement including squint, swallowing difficulty (bulbar signs), opisthotonus, head retroflexion, spasticity, and trismus (pyramidal signs), oculomotor involvement, or generalized tonic-clonic seizures and progressive myoclonic epilepsy in some individuals [6]. Type 1 is the most common form in the European and US populations, with an estimated disease prevalence of 1 in 40,000 to 60,000 individuals in the Czech Republic. Altogether, 62 patients were diagnosed with Gaucher disease between 1975 and 2022. The calculated birth prevalence of Gaucher disease was 1043 patients per 100,000 live births, or 1 patient per 95,853 live births ([7]; data actualized up to March 2023). An increased prevalence is observed in individuals of Ashkenazi Jewish descent. The birth incidence was estimated to be 1 in 450 in the Ashkenazi Jewish population [8], while disease prevalence reaches about 1 in 800 [9,10]. More than 600 *GBA* gene mutations associated with GD have been identified (HGMD^®^ Professional 2022.4).

Guidelines for the evaluation of the disease burden and management of patients with GD can be challenging due to a high degree of phenotypic heterogeneity. The clinical picture can vary from patients remaining asymptomatic for decades to children with severe clinical manifestations [11]. Traditional GD evaluation includes indirect markers (liver and spleen size/volume, hemoglobin level, platelet count, and the skeletal system’s radiologic and MRI characteristics). A personalized guiding treatment decision should be based on pathologically relevant markers that are ideally specific to GD, play a direct role in GD pathophysiology, do not exhibit high genetic variation, reflect the therapy, and can be used for all patients [2]. Monitoring of the disease progression in GD patients is performed at baseline and usually every 3–6 months or in a 12-month period for both treated and untreated patients.

The first and most widely used biomarker is chitotriosidase (CHITO), an orthologue of the chitinase family. While CHITO has been used as an established biomarker since 1994 [12], some concerns have arisen from its description to the present. CHITO limitations include the increased marker activity in different pathological processes [13]. Moreover, approximately 6% of GD Caucasian patients exhibit no measurable CHITO activity due to null alleles in the encoding gene [14], while the manifestation of GD symptoms is not altered in them. Another one-third of GD patients are heterozygotes for this *CHIT1* SNP and thus present with half-normal serum levels [14]. Therefore, evaluating the trend of CHITO activity is more reliable than assessing its absolute levels.

The other alternative biomarkers are also used, irrespective of the chitotriosidase genotype. Lyso-Gb1 concentration from a dried blood spot has been the most specific and sensitive diagnostic GD biomarker [15]. The elevated levels of lyso-Gb1 in GD patients were first shown over 40 years ago [16]. The high lyso-Gb1 levels seem to be related to immune dysregulation and skeletal disease in GD1 patients [17]. Plasma levels of lyso-Gb1 decreased following enzyme replacement therapy (ERT) and were correlated with other biomarker changes [18]. Both CHITO and Lyso-Gb1 reflect the context of GD. They fluctuate in the same direction and can serve as a prognostic and disease-monitoring biomarker in GD [2].

The concentrations of another plasma biomarker, tartrate-resistant acid phosphatase (TRAP), rise with GD progression and decrease in response to ERT. The TRAP isoenzyme 5b is a specific marker of bone resorption. It plays a key role in the degradation of type I collagen by the osteoclast. Like CCL18/PARC, elevation of TRAP activity also appears in other diseases (Niemann–Pick disease, osteopetrosis, and multiple myeloma, among others) [19].

Appropriate candidate biomarker selection is complicated due to the vast clinical and biochemical heterogeneity in GD. Some plasma biomarkers for disease monitoring and response-to-treatment evaluation were or are used, such as ferritin and hemoglobin levels, angiotensin-converting enzyme (ACE), and alkaline phosphatase activity (ALP). However, they were also shown to be affected by other factors, and their lack of specificity was reported [20,21].

The aim of our present study was to test the utility of relevant plasma biomarkers for disease trend evaluation in patients receiving ERT or substrate reduction therapy (SRT) in the Czech cohort of patients with GD.

## 2. Results

### 2.1. Study Population Description

Thirty-four patients with confirmed GD treated with ERT or SRT were included in the present study. All subjects were evaluated during five visits over 30 months. Thirty patients received long-term ERT with imiglucerase (Cerezyme^®^, Sanofi Genzyme, Cambridge, MA, USA) (n = 21) or velaglucerase alfa (VPRIV^®^, Shire Human Genetic Therapies, Inc., MA, USA) (n = 9) during their first study visit. On visit 2, one patient was switched from VPRIV^®^ to Cerdelga^®^ (Sanofi Genzyme, Cambridge, MA, USA). The remaining 29 patients continued ERT during subsequent visits. Three patients started the study period with Cerdelga^®^ and one with Zavesca^®^ (Actelion, Allschwil, Switzerland) (Figure 1). Due to the small SRT sample size, patients with ERT could not be matched to those with SRT. Clinical assessments included CHITO and TRAP activity, platelet counts, and Lyso-Gb1 level analysis at every visit. Baseline characteristics of the study cohort are summarized in Table 1 and Table 2.

**Table 1 ijms-24-14440-t001:** Baseline characteristics of the study cohort. ERT—enzyme replacement therapy; SRT—substrate reduction therapy; CHITO—chitotriosidase; Lyso-Gb1—glucosylsphingosine; TRAP—tartrate resistant acid phosphatase; platelets—platelet count. Values represent the mean ± S.E.M.

Gender	Males: 11 (31.4%)		Females: 23 (68.6%)	
Age (mean, yrs)	42.9 (3–69)			
Genotype	Asn409Ser/Asn409Ser: 1 (2.9%)	Asn409Ser/Leu483Pro: 9 (26.5%)		Asn409Ser/Other or Leu483Pro/Leu483Pro: 24 (70.6%)

Splenectomy	3 (8.6%)			
Therapy	ERT (n = 29)		SRT (n = 5)	
CHITO (nmol/h/mL)	608.3 ± 214.8		4846.0 ± 1638.0	
Lyso-Gb1 (ng/mL)	81.5 ± 12.4		454.9 ± 168.0	
TRAP (nkat/L)	60.9 ± 3.5		124.4 ± 18.1	
Platelets (10^9^/L)	203.9 ± 8.8		116.6 ± 26.7	

**Table 2 ijms-24-14440-t002:** Basic characteristics of the subjects. Pathogenic mutations in the *GBA* and *CHIT1* genes and treatment at initial and follow-up visits are noted. EOW—every other week; mg/D—mg per day.

Patient No.	Gender	Age (Years)	*GBA* Genotype (NM_001005741.3; NP_001005741.1) $	*CHIT1* Genotype (NM_003465.3)	Initial Visit	Follow-Up
1	M	37	p.[Asn409Ser];[Leu483Pro]	p.[Gly102Ser];[=]	Cerdelga^®^ 168 mg/D	Cerdelga^®^ 168 mg/D
2	F	54	p.[Asn409Ser]; [Asp448His;Leu483Pro; Ala495Pro;Val499Val]	p.[Gly102Ser];[=]	Cerezyme^®^ 24 U/kg/EOW	Cerezyme^®^ 26 U/kg/EOW
3	F	41	p.[Asn409Ser]; c.[1265_1319del55]	c.[1049_1072dup];[=]	Cerezyme^®^ 20 U/kg/EOW	Cerezyme^®^ 19 U/kg/EOW
4	M	59	p.[Asn409Ser];[Leu483Pro]	p.[Gly102Ser];[=]	Cerdelga^®^ 168 mg/D	Cerdelga^®^ 168 mg/D
5	M	10	p.[Asn409Ser];[Asp448His]	p.[Gly102Ser];[=]	VPRIV^®^ 50 U/kg/EOW	VPRIV^®^ 39 U/kg/EOW
6	F	47	p.[Asn409Ser];[Leu483Pro]	c.1049_1072dup(;) p.Gly102Ser	Cerezyme^®^ 19 U/kg/EOW	Cerezyme^®^ 19 U/kg/EOW
7	M	47	p.[Asn409Ser]; [Asp448His;Leu483Pro; Ala495Pro;Val499Val]	WT	Cerezyme^®^ 24 U/kg/EOW	Cerezyme^®^ 22 U/kg/EOW
8	M	10	p.[His312Asp];[Asn409Ser]	c.[1049_1072dup];[=]	VPRIV^®^ 61 U/kg/EOW	VPRIV^®^ 42 U/kg/EOW
9	F	3	p.[Arg398Term];[Asn409Ser]	N/D	VPRIV^®^ 32 U/kg/EOW	VPRIV^®^ 24 U/kg/EOW
10	F	25	p.[Arg87Gln];[Asn409Ser]	p.[Gly102Ser];[=]	Cerezyme^®^ 22 U/kg/EOW	Cerezyme^®^ 21 U/kg/EOW
11	F	63	p.[Asn409Ser]; [Leu483Pro;Ala495Pro;Val499Val]	N/D	VPRIV^®^ 31 U/kg/EOW	VPRIV^®^ 29 U/kg/EOW
12 *	F	69	p.[Asn409Ser];[Arg202Term]	p.[Gly102Ser]; [Gly102Ser]	Cerezyme^®^ 41 U/kg/EOW	Cerezyme^®^ 44 U/kg/EOW
13	F	53	p.[Asn409Ser];[Leu483Pro]	p.[Gly102Ser]; [Gly102Ser](;)Ala442Gly (;)Pro451Ser	Cerezyme^®^ 25 U/kg/EOW	Cerezyme^®^ 25 U/kg/EOW
14	F	49	p.[Asn409Ser];[Leu483Pro]	c.1049_1072dup(;) p.Gly102Ser(;) p.Ala442Gly	Cerezyme^®^ 33 U/kg/EOW	Cerezyme^®^ 29 U/kg/EOW
15	M	29	p.[ Asn409Ser];c.[115+1G>A]	WT	Cerezyme^®^ 24 U/kg/EOW	Cerezyme^®^ 22 U/kg/EOW
16	M	33	p.[Asn409Ser];[Leu483Pro]	WT	Cerezyme^®^ 18 U/kg/EOW	Cerezyme^®^ 18 U/kg/EOW
17 *	F	59	p.[Asn409Ser];[Leu483Pro]	WT	Cerezyme^®^ 26 U/kg/EOW	Cerezyme^®^ 26 U/kg/EOW
18	M	28	p.[Asn409Ser];[Gly416Ser]	p.Gly102Ser(;) p.Ala442Gly	Cerezyme^®^ 17 U/kg/EOW	Cerezyme^®^ 17 U/kg/EOW
19 ^#^	M	32	p.[Leu483Pro];[Leu483Pro]	c.[1049_1072dup];[=]	Cerezyme^®^ 41 U/kg/EOW	Cerezyme^®^ 44 U/kg/EOW
20	F	31	p.[Asn409Ser];[Leu483Pro, Val499Val]	c.[1049_1072dup];[=]	Cerezyme^®^ 24 U/kg/EOW	Cerezyme^®^ 25 U/kg/EOW
21	M	42	p.[Asn409Ser];c.[1326dupT]	p.[Gly102Ser];[=]	VPRIV^®^ 28 U/kg/EOW	Cerdelga^®^ 28 U/kg/EOW
22 *	F	56	p.[Asn409Ser]; c.[1265_1319del55]	WT	VPRIV^®^ 23 U/kg/EOW	VPRIV^®^ 22 U/kg/EOW
23	F	66	p.[ Asn409Ser];[Gly241Glu]	p.[Gly102Ser];[=]	Cerezyme^®^ 15 U/kg/EOW	Cerezyme^®^ 15 U/kg/EOW
24	F	19	c. [Gly228Term]; p.[Asn409Ser]	N/D	Cerezyme^®^ 40 U/kg/EOW	Cerezyme^®^ 37 U/kg/EOW
25	F	57	p.[Asn409Ser];[Leu483Pro; Ala495Pro;Val499Val]	c.[1049_1072dup];[=]	VPRIV^®^ 26 U/kg/EOW	VPRIV^®^ 25 U/kg/EOW
26	F	33	p.[Asn409Ser];[Ser235Pro]	p.[Gly102Ser];[=]	Cerezyme^®^ 18 U/kg/EOW	Cerezyme^®^ 16 U/kg/EOW
27	F	46	p.[Asn409Ser];[Leu483Pro; Ala495Pro;Val499Val]	p.[Gly102Ser];[=]	Zavesca^®^ 200 mg/D	Zavesca^®^ 200 mg/D
28	F	59	p.[ Asn409Ser];c.[115+1G>A]	c.[1049_1072dup];[=]	VPRIV^®^ 28 U/kg/EOW	VPRIV^®^ 25 U/kg/EOW
29	F	69	p.[Asn409Ser];[Leu483Pro; Ala495Pro;Val499Val]	c.[1049_1072dup];[=]	Cerezyme^®^ 34 U/kg/EOW	Cerezyme^®^ 35 U/kg/EOW
30	M	45	p.[Asn409Ser]; c.[1265_1319del55]	c.[1049_1072dup];[=]	VPRIV^®^ 28 U/kg/EOW	VPRIV^®^ 29 U/kg/EOW
31	F	45	p.[Asn409Ser];[Leu483Pro; Ala495Pro;Val499Val]	p.[Gly102Ser];[=]	Cerezyme^®^ 28 U/kg/EOW	Cerezyme^®^ 29 U/kg/EOW
32	F	46	p.[Arg398Gln ];[Asn409Ser]	WT	Cerezyme^®^ 23 U/kg/EOW	Cerezyme^®^ 28 U/kg/EOW
33	F	39	p.[Asn409Ser];[Leu483Pro]	p.[Gly102Ser];[=]	Cerdelga^®^ 164 mg/D	Cerdelga^®^ 164 mg/D
34	F	54	p.[Asn409Ser];[Asn409Ser]	p.[Ser308Ile];[=]	Cerezyme^®^ 26 U/kg/EOW	Cerezyme^®^ 24 U/kg/EOW

M, male; F, female; ERT, enzyme replacement therapy; SRT, substrate reduction therapy. VPRIV^®^, ERT with velaglucerase alfa; Cerezyme^®^, ERT with imiglucerase; SRT with eliglustat, Cerdelga^®^; SRT with miglustat, Zavesca^®^; *, splenectomy; #, patient of GD III phenotype; inf, infusion; N/D, not done; $ The set of amino acid substitutions Asp448His; Leu483Pro; Ala495Pro; Val499Val in patients 2 and 7 and Leu483Pro; Ala495Pro; Val499Val in patients 11, 20, 25, 27, 29, 31 is a consequence of recombination event between *GBA* and *GBA* pseudogene and corresponds to rec(g.4889–6506) and recNciI, respectively, as specified in (Hodanova et al., 1999 [22]).

The effect of ERT and SRT was assessed with changes in Lyso-Gb1 levels, CHITO and TRAP activities, and platelet count in a mostly female study cohort of 34 patients. At Visit 0, 30 of the included patients were treated with ERT and four with SRT (Table 2). On visit 2, one patient was switched from ERT to SRT (Figure 1, Table 2, no. 21). No patient interrupted treatment. Prior to entering the study, patients on ERT therapy had been treated for 15.6 years on average (range 2.0 to 25.0) and patients on SRT for 2.8 years (range 0.0 to 11 years), respectively. At the time of the first visit, the average age was 42.9 years (range 3–69 years).

A genotype–phenotype correlation was clearly observed. In thirty-three patients exhibiting phenotype I GD, the presence of at least one p.Asn409Ser allele was detected (32 patients were heterozygous for this allele; patient no. 34 was a homozygote). Patient no. 19, homoallelic for p.Leu483Pro, presented with phenotype III GD (Table 2). Overall, 3 (8.9%) patients were splenectomized (Table 1). 

The chitotriosidase genotype was determined in 30 out of 34 patients involved in the study. Ten patients were heterozygous carriers of the dup24 allele (c.1049_1072dup; rs3831317) in the *CHIT1* gene. In the remaining 20 patients, the dup24 allele was not detected. Three other known single nucleotide variants, c.304G>A (p.Gly102Ser; rs2297950), c.1325C>G (p.Ala442Gly; rs1065761), c.1351C>T (p.Pro451Ser; rs141079733), and one unpublished variant, c.922_923delinsAT (p.Ser308Ile), were detected within the cohort (Table 2).

### 2.2. Biomarker Evolution during Study Period

At every scheduled visit, a peripheral blood sample was drawn for CHITO and TRAP activity, Lyso-Gb1 levels, and platelet count assessment. The results of biomarker changes at the beginning and end of the study period are summarized in Table 3. Figure 2 depicts plasma marker evolution in both groups during the study period.

### 2.3. Response of CHITO Activity to Long-Term Therapy

To assess the impact of long-term ERT and SRT on CHITO activity, we measured the serum activity of CHITO at regular 6-month intervals. After 30 months, the reduction in CHITO activity was observed irrespective of treatment and chitotriosidase genotype. After ERT treatment, plasma CHIT activity decreased from a median of 608.3 ± 214.8 nmol/h/mL to 244.7 ± 52.6 nmol/h/mL (*p* = 0.123, Table 3, Figure 2A). After SRT treatment, we observed a marked decrease in plasma CHITO activity from a median of 4846.0 ± 1638.0 nmol/h/mL to a median of 1780.0 ± 743.0 nmol/h/mL (*p* = 0.045, Table 3, Figure 2E). Linear regression analysis was performed, and a regression equation was calculated. Each point of the regression curve corresponded to the mean of values measured at regular 6-month visits, giving a sample size of 5. For ERT treatment, the regression curve showed a slope value ± standard error of −9.68 ± 2.27 (Figure 3A). We observed a more skewed regression curve for SRT treatment with a slope value ± standard error of −106.64 ± 18.11 (Figure 3E). Further, we compared the slopes of linear regression curves to determine whether the slopes of the two treatments were significantly different from each other, given the slope, standard error, and sample size for each line. The two slopes differed significantly (*p* < 0.001). The isotonic regression curve was estimated using the pool adjacent violators algorithm and fitted to minimize the mean squared error (Figure 4A). The patients treated with SRT demonstrated a sharper decline in CHITO activity.

### 2.4. Response of Lyso-Gb1 Levels to Long-Term Therapy

To compare Lyso-Gb1 responses to long-term therapy, we estimated Lyso-Gb1 levels over 30 months of treatment. The median Lyso-Gb1 level dropped comparably, irrespective of treatment. By 30 months of ERT treatment, there was a significant decrease in plasma Lyso-Gb1 level from a median of 81.5 ± 12.4 ng/mL to 60.9 ± 8.6 ng/mL (*p* = 0.048, Table 3, Figure 1B). After SRT treatment, we observed a marked drop in plasma Lyso-Gb1 levels from a median of 454.9 ± 168.0 ng/mL to a median of 114.4 ± 45.7 ng/mL (*p* = 0.002, Table 3, Figure 2F). For ERT treatment, the Lyso-Gb1 regression curve showed a slope value ± standard error of −0.849 ± 0.46 (Figure 2B). We observed a more skewed regression curve for SRT treatment with slope value ± standard error −7.976 ± 4.75 (Figure 3F). Even though the patients treated with SRT demonstrated a sharper decline in Lyso-Gb1 levels, slope comparison revealed comparable rate behavior of the marker decrease, irrespective of treatment (*p* < 0.17). The isotonic regression curves for both treatments are shown in Figure 4B.

### 2.5. Response of TRAP Activity to Long Term Therapy

To analyze whether ERT resulted in significant differences in serum TRAP activity compared to SRT, we evaluated median TRAP activities. At month 30, an overall increase in median TRAP activity values was observed within the ERT cohort, whereas median TRAP activity dropped in the subjects treated with SRT. Both trends were significant for the particular group. The median Lyso-Gb1 level dropped irrespective of treatment. In the ERT patients, we observed a significant increase in plasma TRAP activity from a median of 60.9 ± 3.5 nkat/L to 73.5 ± 3.1 nkat/L (*p* = 0.001, Table 3, Figure 2C). The regression curve equation showed a positive slope value (slope ± standard error) of 0.419 ± 1.01 (Figure 3C). Within the SRT group, there was an initial, marked drop in TRAP activity, followed by trend stagnation. The median TRAP activity value decreased from a median of 124.4 ± 18.1 nkat/L to a median of 95.8 ± 11.1 nkat/L (*p* = 0.041, Table 3, Figure 2G). We observed a downtrend regression curve for SRT treatment with a slope value ± standard error of −1.055 ± 0.46 (Figure 3G). The two slopes differed significantly (*p* < 0.015). Our data suggests that long-term SRT could affect bone resorption in GD I patients more effectively than ERT. The isotonic regression curves for both treatments are depicted in Figure 4C.

### 2.6. Response of Platelet Count to Long-Term Therapy

The platelet count recovered steadily and significantly over 30 months, regardless of the treatment method. Compared to ERT, the effect of the therapy on platelet count increase was more noticeable in patients treated with SRT. By 30 months of ERT treatment, there was a significant increase in platelet count from a median of 203.9 ± 8.8 × 10^9^/L to 229.0 ± 12.4 × 10^9^/L (*p* = 0.006, Table 3, Figure 2D). In patients with SRT, the median platelet count increased from a median of 116.6 ± 26.7 × 10^9^/L to a median of 196.0 ± 28.6 × 10^9^/L (*p* = 0.028, Table 3, Figure 2H). Both regression curves depicted the same trend, with a positive slope skewed significantly more in the SRT cohort than in the ERT cohort. The platelet count regression curve for ERT treatment showed a slope value ± standard error of 0.7417 ± 0.19 (Figure 3D). For SRT treatment, we observed a regression curve with a slope value ± standard error of 2.4457 ± 0.50 (Figure 3H). Slope comparison revealed a significant difference in the increase rate between the treatment cohorts (*p* < 0.012). The isotonic regression curve revealed a continuously increasing trend (Figure 4D).

### 2.7. Correlation of Lyso-Gb1 Levels with Other Biomarkers

To test the utility of Lyso-Gb1 for monitoring response in long-term treatment patients with GD, Lyso-Gb1 was correlated with established serum markers of Gaucher disease. Plasma lyso-GL1 correlated significantly with chitotriosidase (ERT: r = 0.54, *p* < 0.001; SRT: r = 0.83, *p* < 0.001) and TRAP (ERT: r = 0.34, *p* < 0.001; SRT: r = 0.88, *p* < 0.001) irrespective of treatment method (Table 3, Figure 5A,B,D,E). There was no significant correlation between Lyso-Gb1 and platelet count within the ERT cohort (r = −0.01, *p* = 0.92) (Table 3, Figure 5C). On the contrary, patients receiving SRT showed a significant correlation between Lyso-Gb1 and platelet count (r = −0.45, *p* = 0.0249) (Table 3, Figure 5F). The association of Lyso-Gb1 with all evaluated markers was stronger in the SRT cohort. 

## 3. Discussion

The system-wide involvement of Gaucher disease requires regular monitoring of all GD patients. Appropriate biomarkers correlating with the disease burden are important for monitoring disease progression and response-to-treatment evaluation. However, the choice of the optimal biomarker has its limitations, reflecting their lack of specificity, different sensitivity and specificity, or the fact that they can be affected by other factors [20,21]. Recent studies reported Lyso-Gb1 as the most specific and sensitive diagnostic and disease-monitoring biomarker with the potential to be a prognostic biomarker of GD [2,13,23,24,25,26]. Our study assessed the utility of pathophysiologically relevant plasma biomarker combinations for disease trend evaluation in patients receiving long-term ERT or SRT. 

Our investigation confirmed the utility of CHITO in monitoring disease progression and response to therapy. Indeed, CHITO usage in monitoring treatment efficacy was credited to a substantial body of evidence [27,28,29,30,31,32]. Our results revealed that in patients with long-term therapy (the mean number of years in treatment in the ERT cohort was 15.6 years), a decline in CHITO levels is still evident. However, it reached statistical significance with the skewer slope of the regression curve only within the SRT cohort (the mean of the previous treatment was 2.8 years). We attribute the lower CHITO decrease rate in the ERT cohort mainly to the effect of time; Hollak [27] reported a decrease in chitotriosidase activity of 32% in 1 year, and Vigan and colleagues [33] found in their predictive model based on data from 233 GD patients a 95% CHITO response in 2 years and a 36% response in 1 year with ERT. On the contrary, our patients have yet to reach some sustainable plateau since the start of their ERT/SRT, resulting in a less marked decrease in CHITO activity. The higher decrease rate within the SRT cohort could also reflect significantly higher baseline (at month 0 of the study) CHITO activities in SRT patients. Patients with severe changes in the disease’s biomarker levels could profit more from therapy and show more marked changes in biomarker levels or activities. The *GBA* gene mutation heterogeneity was not equal between our study cohorts (Table 2). Within a limited number of our SRT population, 3 of 5 (60%) patients were of the p.[Asn409Ser];[Leu483Pro] genotype, and one patient (no. 21) harbored a null mutation c.1326dupT, which may be considered even more severe than the Leu483Pro genotype Rec Ncil2 mutation, which is considered even more severe than the Leu483Pro genotype [34]. Other factors one should bear in mind are the different mechanisms of SRT and ERT action, as well as the different mechanisms of eliglustat and miglustat as compared to ERT (as undermentioned). 

The results of *CHIT1* genotype determination among our Gaucher patients and estimation of CHITO activity in heterozygotes for the dup24 allele were consistent with the reports by the other groups [35,36,37,38]; however, we detected no dup24 and two p.Gly102Ser homozygotes among the same patients. As reported by Bussing and colleagues [38], the concomitant presence of the p.Gly102Ser allele may result in an underestimation of disease severity if evaluated by CHITO activity. We did not dispose of a priori data on whether dup24 and p.Gly102Ser mutations are at the same or distinct *CHIT1* alleles in our patients. However, we did not correct CHITO activity by a factor of 1.6 in our two p.Gly102Ser patients, as recommended by Bussink et al. [38]. Again, the insights of our study are admittedly limited by the small number of patients.

We further analyzed changes in Lyso-Gb1 levels, presently the superior diagnostic biomarker for GD [18,23]. Rolfs and colleagues [23] estimated a plasma Lyso-Gb1 level of 12 ng/mL (25.99 nmol/mL) as a threshold differentiating patients with GD from healthy subjects or patients with other lysosomal storage, with 100% sensitivity and 100% specificity. In a comprehensive review of Revel-Vilk et al. [21], data from 16 of the 17 studies demonstrated that treatment with ERT and SRT (alone or in combination) led to marked reductions in Lyso-Gb1 plasma levels, cerebrospinal fluid (CSF), and urine [21]. Lyso-Gb1 was also shown to precede and predict the response of splenomegaly and thrombocytopenia in patients with ERT [39]. Our results are in concord with these studies. A decrease in Lyso-Gb1 levels following ERT or SRT in our GD patients proved comparable rate behavior for the marker decrease, irrespective of treatment. As distinct from the CHITO evaluation, regression curve slopes did not differ significantly. Lyso-Gb-1 levels are not hampered by the well-described limitations of CHITO activity [38]. Even though Lyso-Gb1 levels were significantly higher in SRT patients at Month 0, the trendlines of both ERT and SRT long-term patients were closely similar. That is in good agreement with observations suggesting that a significant reduction in plasma lyso-Gb1 occurs after ERT or SRT initiation before stabilizing at a lower concentration in most patients [23,40,41].

Interestingly, at month 30, an overall significant increase in median TRAP activity values was observed within the ERT cohort. In contrast, median TRAP activity dropped (with borderline significance) in the subjects treated with SRT in our study. The isotonic regression curve depicted reached a plateau in TRAP activity decrease in patients with SRT treatment, whereas an increasing trend was evident in the ERT group.

It was documented that the TRAP level could differ as a function of age [42]; however, the respective age means weren’t significantly different between our ERT and SRT patients. Considering biomarker values, the disease burden, including bone involvement, was significantly higher within our SRT cohort. TRAP activity rises during bone resorption through the degradation of type I collagen by the osteoclast. We can only hypothesize that higher TRAP levels will better respond to the particular treatment. Sims and colleagues [43] evaluated changes in bone disease in 33 patients with skeletal manifestations with ERT therapy with imiglucerase. They studied bone formation (osteocalcin, bone-specific alkaline phosphatase) and bone resorption (N-telopeptide crosslinks and deoxypyridinoline, D-PYD) markers. With median baseline measurements of all markers at baseline distributed within the normal range, median post-treatment measurements for bone formation markers were significantly higher, even though some increases were small, and all increased within the normal reference ranges. The decrease in bone resorption markers was not significant; however, their results indicate an alteration favoring new bone formation relative to bone resorption after 48 months with ERT treatment [43]. The effect of SRT on bone involvement in GD seems more complex. Miglustat, acting as a competitive inhibitor of glucosylceramide synthase, blocks the first committed step in glycosphingolipid synthesis, thus blocking the synthesis of all glucosylceramide-derived glycosphingolipids. However, it was shown to have dual mechanisms of action. It may also accelerate the degradation of the glycolipid complex by increasing glucocerebrosidase activity, which can be additionally beneficial in patients harboring mutations with residual enzyme activity (such as p.Asn409Ser) [44]. Moreover, in a study using human and mouse NPC1-mutant cells, miglustat demonstrated its ability to affect impaired calcium homeostasis related to sphingosine storage [45]. The second SRT, eliglustat, is a highly specific oral glucosylceramide synthase inhibitor structurally different from miglustat, upon which eliglustat exhibits superior potency and selectivity [46]. The eliglustat effect is also weighty on bone involvement in GD. Eliglustat prevents autophagic degradation of TNF receptor-associated factor 3 (TRAF3), a key step in osteoclast differentiation [47]. This SRT effectively inhibits autophagy in osteoclasts and increases bone mass, thus having tremendous potential for the therapeutic use of eliglustat in bone loss. Taken together, our data could indicate a marked reduction in osteoclastogenic markers in patients with SRT treatment compared to the ERT cohort. Nevertheless, the markers of bone metabolism and their level patterns are complex. Their variability makes their use in clinical practice rather polemical [19,48,49].

The platelet count recovered steadily and significantly over 30 months, regardless of the method of treatment. The trends of platelet count recovery were comparable for both therapies, and our observation confirms literary evidence [43,48,50]. Mistry et al. [51] showed improvement in cytopenias with different dosing regimens of ERT. In slow or poor responders, a gradual increase in the ERT dose can increase the platelet count; the other method of choice is to switch to another ERT regimen. Elstein et al. [52] demonstrated approximately a 40% chance of improvement (“booster effect”) due to a switch from imiglucerase or taliglucerase alfa to velaglucerase alfa. A possible therapeutic benefit of SRT was demonstrated in patients with poor platelet response after 6 years on first-line ERT treatment. After ERT discontinuation, eliglustat was the sole treatment, leading to a good clinical and relatively stable platelet response [53]. The study also illustrates GD’s different responses to ERT or SRT treatment [50].

In our ERT and SRT study cohorts, we observed a significant correlation of Lyso-Gb1 with CHITO and TRAP markers. A platelet count increase was associated with a Lyso-Gb1 decrease in both treatment groups; however, the correlation was insignificant in ERT patients. The association of Lyso-Gb1 with all evaluated markers was markedly stronger in the SRT cohort. Correlations between Lyso-Gb1 levels and other established biomarkers of GD were summarized in several studies [2,18,54,55,56]. Significant correlations between plasma Lyso-Gb1 and CHITO activity were demonstrated, with Pearson coefficient values varying from 0.59 to >0.9 [18,54,55,56]. These findings are in good agreement with our results.

With respect to correlations, there is evidence of a moderate pairing between decreasing plasma Lyso-Gb1 levels and increasing platelet counts in patients with ERT [48,52]. In the present study, we found no significant correlation between Lyso-Gb1 and platelet count within the ERT cohort. On the contrary, patients receiving SRT showed a significant correlation between Lyso-Gb1 and platelet count (r = −0.45). In the paper of Hurwitz and colleagues [25], Lyso-Gb1 significantly correlated with platelet count in patients with ERT, with almost the same correlation coefficient of −0.42 [25]. On the other hand, although platelet count increased with Lyso-Gb1 decrease in the pooled ERT and SRT cohort of 128 patients in the study of Murugesan et al. [2], no correlation between Lyso-Gb1 and platelet counts was found. As tartrate-resistant acid phosphatase is now included in the older and less specific GD markers, the limited literary evidence of Lyso-Gb1 and TRAP correlation in GD patients hinders the formation of further conclusions about its significance. However, regarding all tested correlations, we found the highest correlation coefficient for the association of Lyso-Gb1 and TRAP in patients with SRT treatment. With respect to the above-mentioned combined effects of SRT on bone involvement in GD patients, Lyso-Gb1 correlation with TRAP could still be considered, at least for SRT therapy response monitoring by means of osteoclast activity, as a cheap and available instrument.

The present study has several limitations, and the results should be interpreted with caution. First, our study sample is of a very limited size. Moreover, the ERT to SRT patient ratio is highly inequal (6 to 1) and differs in the duration of treatment before entering the study. One of the further limitations of the study was the inclusion of a higher ratio of females to males (24F/11M), reflecting our center’s patient population. Another limitation of the study was the heterogeneity of patients in treatment dosage and the wide variation in the *GBA1* mutation. Regardless, our data confirmed the effectiveness of ERT and SRT long-term treatment in patients with GD. We proved the non-inferiority of SRT. Our results indicate that ERT and SRT show the potential of individual drug effectiveness in treating particular symptoms, either as a combination or in a switch manner.

GD has possibly the widest range of thoroughly studied biomarkers among lysosomal storage disorders. Combining the key biomarker of GD and the evaluation of the treatment, Lyso-Gb1, with other biomarkers, can offer an improved assessment of response to long-term therapy. Nevertheless, new therapeutic options and reliable biomarkers for GD monitoring are crucial to tailoring therapies to a patient’s specific clinical needs.

## 4. Materials and Methods

### 4.1. Biochemical Plasma Markers

#### 4.1.1. CHITO Activity

Chitotriosidase activity in plasma samples stored at −20 °C was determined using the fluorogenic substrate 4-methylumbelliferyl-β-D-*N*-*N*′-*N*″-triacetylchitotriose (Sigma M-5639, Steinheim, Germany). The assay was performed according to the Hollak method. Fluorescence of 4-methylumbelliferone released from the substrate was detected at excitation 365 nm and emission 448 nm by a luminescence spectrometer (Perkin Elmer LS50B, Wellesley, MA, USA).

#### 4.1.2. Chit-1 Genotyping

Genomic DNA was isolated from peripheral blood leukocytes using the QIAamp DNA Blood Mini Kit (Qiagen, Valencia, CA, USA). Purified DNA of the entire coding region of the 12-exon human CHIT1 gene, including intron/exon junctions (GenBank NC_000001.11, NM_003465.3), was used for PCR amplification. Each amplicon was sequenced by Sanger sequencing or by high-throughput sequencing using the NexteraXT kit and the MiSeq platform (Illumina, San Diego, CA, USA).

#### 4.1.3. Lyso-Gb1

Tandem mass spectrometry from dried blood spots was used to quantitatively measure Lyso-Gb1. Blood samples were collected as dried blood spots on filter cards. Lyso-Gb1 levels were performed using liquid chromatography-mass spectrometry at ARCHIMEDlife Laboratories (ARCHIMEDlife Laboratories, Vienna, Austria).

#### 4.1.4. TRAP Activity

Serum TRAP activity was measured by a spectrophotometric assay in the presence of 0.1 M L(+) sodium tartrate at pH 5.6 using 10 mM 4-nitro-phenyl phosphate as substrate.

### 4.2. Statistical Analysis

Descriptive analysis and a test for the characteristics of a normal distribution (the Kolmogorov–Smirnov test of normality) were performed for sample background characteristics. Continuous variables were presented as a range of values as minimum and maximum, mean and standard error of the mean (SEM), or median and standard deviation (SD), depending on the distribution. A paired sample *t*-test was used to assess differences in plasma markers over time. Two sample *T*-tests were performed to test differences between therapies for statistical significance. To describe the relationships between the effects of the treatment and time, a linear regression analysis was performed and a regression equation was calculated. Each point of the regression curve corresponded to the mean of values measured at regular 6-month visits, giving a sample size of 6. Slopes of linear regression curves were compared to determine whether the slopes of the two treatments were significantly different from each other, given the slope, standard error, and sample size for each line (c.f. [57]). The isotonic regression curve was estimated using the pool adjacent violators algorithm and fitted to minimize the mean squared error. All *p*-values reported are two-tailed, and *p*-values < 0.05 were considered significant. In univariate analysis, Pearson’s correlation coefficient *r* was used to correlate Lyso-Gb1 against CHITO, TRAP, and platelet count. Statistical analysis was performed using MedCalc version 20.010 (MedCalc Software Ltd., Ostend, Belgium) and OriginPro 8.5.0 SR1 (OriginLab Corporation, Northampton, MA, USA).

## Figures and Tables

**Figure 1 ijms-24-14440-f001:**
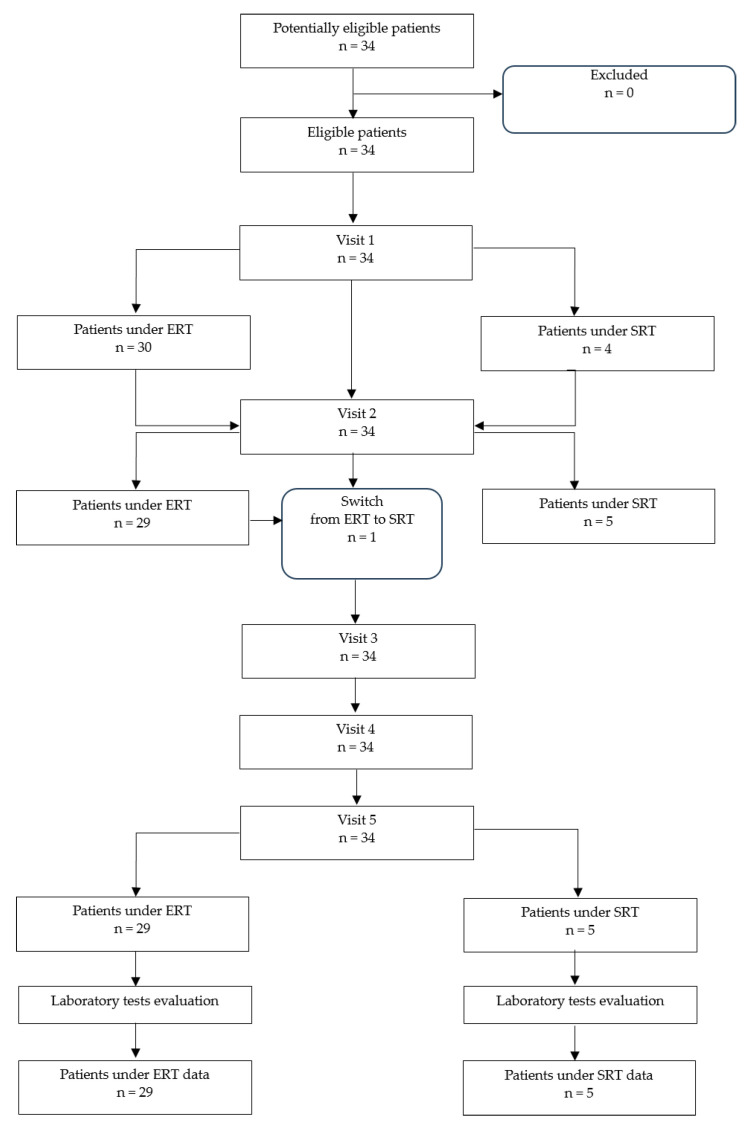
Study design flowchart.

**Figure 2 ijms-24-14440-f002:**
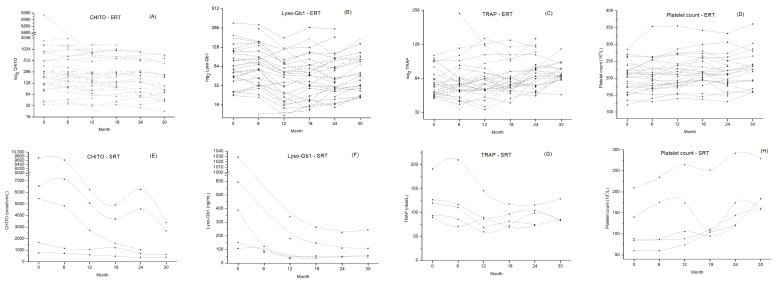
Plasma markers’ evolution during the study period. Upper panels (graphs (**A**–**D**)) represent absolute marker activities for each subject with ERT. Lower panels (graphs (**E**–**H**)) represent absolute marker activities for each subject with SRT. For patients with ERT, CHITO, Lyso-Gb1, and TRAP were log-transformed to achieve a normal distribution for the graphical presentation. Graphs (**A**,**E**) CHITO activity; Graphs (**B**,**F**) Lyso-Gb1 levels; Graphs (**C**,**G**) TRAP activity; Graphs (**D**,**H**) platelet count. ERT—enzyme replacement therapy; SRT—substrate reduction therapy; CHITO—chitotriosidase; Lyso-Gb1—glucosylsphingosine; TRAP—tartrate-resistant acid phosphatase.

**Figure 3 ijms-24-14440-f003:**
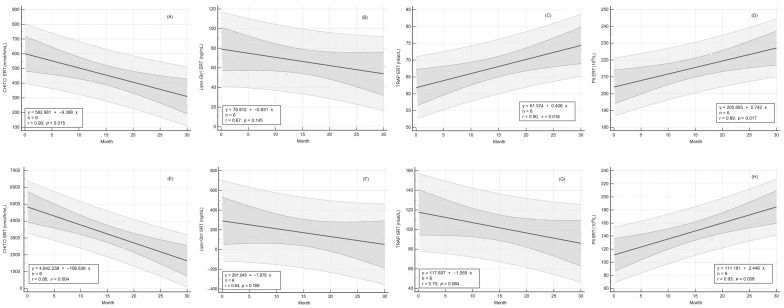
Regression curves of plasma markers. Each graph depicts the regression line, 95% confidence interval (dark gray), the 95% prediction interval (light gray), and the equation of the regression curve (framed). Upper panels (graphs (**A**–**D**)) represent marker regression curves for ERT. Lower panels (graphs (**E**–**H**)) represent regression curves for SRT. Graphs (**A**,**E**) CHITO regression curves; Graphs (**B**,**F**) Lyso-Gb1 regression curves; Graphs (**C**,**G**) TRAP regression curves; Graphs (**D**,**H**) platelet count regression curves. ERT—enzyme replacement therapy; SRT—substrate reduction therapy; CHITO—chitotriosidase; Lyso-Gb1—glucosylsphingosine; TRAP—tartrate-resistant acid phosphatase.

**Figure 4 ijms-24-14440-f004:**
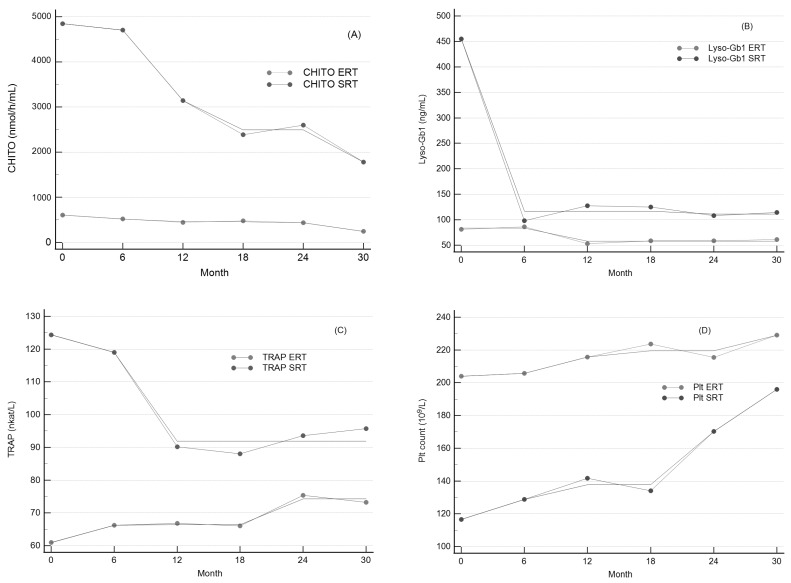
The graph shows changes in (**A**) CHITO activity, (**B**) Lyso-Gb1 levels, (**C**) TRAP activity, and (**D**) platelet count together with the isotonic regression curve (a solid line without markers used to constraint means the line does not decrease). Each point of the regression curve corresponds to the mean of values measured at regular 6-month visits in all graphs.

**Figure 5 ijms-24-14440-f005:**
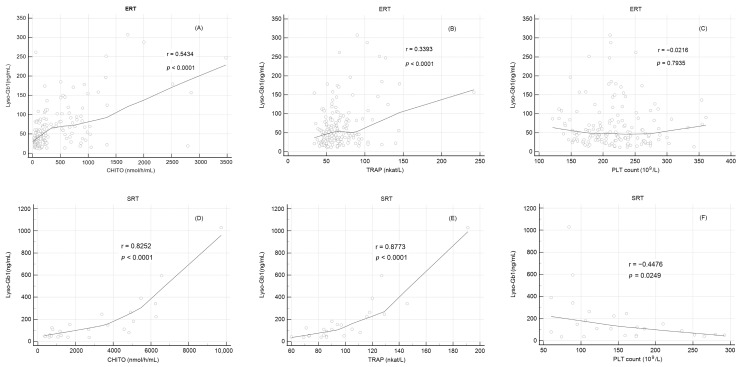
Lyso-Gb1 correlation with evaluated biomarkers over a 30-month period. The upper panel row represents patients with ERT. (**A**) Lyso-Gb1 correlation with CHITO. (**B**) Lyso-Gb1 correlation with TRAP. (**C**) Lyso-Gb1 correlation with platelet count. The lower panel row depicts patients with SRT. (**D**) Lyso-Gb1 correlation with CHITO. (**E**) Lyso-Gb1 correlation with TRAP. (**F**) Lyso-Gb1 correlation with platelet count. CHITO—chitotriosidase; Lyso-Gb1—glucosylsphingosine; TRAP—tartrate-resistant acid phosphatase.

**Table 3 ijms-24-14440-t003:** Median decrease and increase of biomarkers and Lyso-Gb1 correlation with CHITO, TRAP, and platelet count in ERT and SRT patients.

	ERT (n = 29)					SRT (n = 5)				
Duration of Therapy before Entering the Study	15.6 Years (Range 2.0 to 25.0 Years)					2.8 Years (Range 0.0 to 11 Years)				
	Month 0	Month 30		*p*-Value	Trend	Month 0	Month 30		*p*-Value	Trend
CHITO (nmol/h/mL)	608.3 ± 214.8	244.7 ± 52.6		0.123	↓	4846.0 ± 1638.0	1780.0 ± 743.0		0.045	↓
Lyso-Gb1 (ng/mL)	81.5 ± 12.4	60.9 ± 8.6		0.048	↓	454.9 ± 168.0	114.4 ± 45.7		0.002	↓
TRAP (nkat/L)	60.9 ± 3.5	73.5 ± 3.1		0.001	↑	124.4 ± 18.1	95.8 ± 11.1		0.041	↓
Platelet (10^9^/L)	203.9 ± 8.8	229.0 ± 12.4		0.006	↑	116.6 ± 26.7	196.0 ± 28.6		0.028	↑
**Lyso-Gb1 Correlation with CHITO, TRAP, and Platelets Count**										
	ERT (n = 30)					SRT (n = 5)				
	Lyso-Gb1					Lyso-Gb1				
	R		*p*-value			R		*p*-value		
CHITO	0.54		<0.0001			0.83		<0.0001		
TRAP	0.34		<0.0001			0.88		<0.0001		
Platelets	−0.01		0.92			−0.45		0.0249		

ERT—enzyme replacement therapy; SRT—substrate reduction therapy; CHITO—chitotriosidase; Lyso-Gb1—glucosylsphingosine; TRAP—tartrate resistant acid phosphatase; platelets—platelet count.

## Data Availability

The data that support the findings of this study are available from the corresponding author, M.M., upon reasonable request.

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
