# Peer review of "Long-Term Evaluation of Biomarkers in the Czech Cohort of Gaucher Patients"

_ijms, 2023, doi:10.3390/ijms241914440_

Round 1
Reviewer 1 Report
1. The method doesn't explain or justify the time points 0 and 30 to measure the activity of the enzyme.
2. The table to show the gene mutation can be summarised to highlight its usefulness.
Author Response
First of all we would like to thank the reviewer for his/her positive feedback.
- The method doesn't explain or justify the time points 0 and 30 to measure the activity of the enzyme.
The interval of 30 months was determined arbitrary as the data for this time period were available from all patients.
- The table to show the gene mutation can be summarised to highlight its usefulness.
Thank you for your comment. As for the GBA genotype we preferred to show detailed data of all analysed patients in the Table. As the list of GBA variants was not the main topic of the manuscript, we have not commented on this topic anymore. As for the CHIT1 genotype, the results are summarized and commented in the Discussion (lines 297-306).
Reviewer 2 Report
Dear Authors,
This study evaluated the trend of specific Gaucher disease markers such as chitotriosidase, tartrate-resistant acid phosphatase activity, Lyso-Gb1 and platelet count during 30 months of observation according to ERT or SRT therapy. The purpose, as I understand it, was to test these biomarkers for the assessment of disease progress in patients undergoing ERT or substrate reduction therapy (SRT) in this Czech GD cohort.
Comments and criticisms:
Other important disease markers such as organomegaly and bone involvement were not reported in this article. The biochemical parameters analyzed provide similar and highly correlated information, as demonstrated by the analysis itself, therefore the possible evaluation of a disease marker such as the volume of the spleen and liver could rather than bone density have given more strength to these data.
It should be noted in the introduction that neurological involvement is also widely recognized in GD type 1, authors should refer to the neurological involvement of these patients (e.g. Parkinson's disease and movement disorders).
Line 48: It is not specified which is the most common type of GD
In table 1: is it necessary to specify how the dispersion measures of the mean (SD, SE, range) are reported?
Table 3 should also report the duration of treatment for both the ERT and SRT groups: an explanation of the results obtained could be that while the ERT group had been on treatment for a long time and therefore reached a plateau for which it is more difficult to notice changes, the SRT group started therapy more recently, so the 30 months of observation looked at the initial effect of the drug before reaching platou.
In Table 1, did the p-value refer to the differences between ERT and SRT? You should divide this part from the rest of the table. Furthermore, this information is quite imprecise because it seems that SRT was more effective, while this depends on the baseline values of the parameters evaluated, on the age of the patient being treated.
In Table 2, instead of the total dose of ERT per patient, it would be appropriate to report the dose per kg of body weight administered every 2 weeks. The same should be done in the article.
Figures 1 and 2 are not drawn correctly and should be revised to make them more readable.
Your bone metabolism data was mentioned in the discussion (line 351), in this case this data should be shown
Even if the number is low, perhaps the patient who switched from ERT to SRT should be eliminated from the analysis because it is not clear in which group he was analysed.
Minor editing of English language required
Author Response
Dear Authors,
This study evaluated the trend of specific Gaucher disease markers such as chitotriosidase, tartrate-resistant acid phosphatase activity, Lyso-Gb1 and platelet count during 30 months of observation according to ERT or SRT therapy. The purpose, as I understand it, was to test these biomarkers for the assessment of disease progress in patients undergoing ERT or substrate reduction therapy (SRT) in this Czech GD cohort.
We would like to thank the reviewer for the his/her feedback and inspiring comments.
Comments and criticisms:
Other important disease markers such as organomegaly and bone involvement were not reported in this article. The biochemical parameters analyzed provide similar and highly correlated information, as demonstrated by the analysis itself, therefore the possible evaluation of a disease marker such as the volume of the spleen and liver could rather than bone density have given more strength to these data.
It should be noted in the introduction that neurological involvement is also widely recognized in GD type 1, authors should refer to the neurological involvement of these patients (e.g. Parkinson's disease and movement disorders).
Thank you for this remark. We have updated the text with two new references:
Symptoms of the most prevalent type 1 GD may include hepatosplenomegaly, anemia, thrombocytopenia, growth delay, and bone or pulmonary involvement. The majority of heterozygous mutations in the GBA1 gene elevate the risk of Parkinson disease and dementia with Lewy bodies. Even though type 1 GD patients classically do not CNS involvement, they are at increased risk for development of parkinsonism and Parkinson disease (PD). Heterozygous mutations in the GBA1 gene seem to cause a more severe PD phenotype and are associated with synucleinopathies in general (Thaler et al., 2017). However, large-scale studies report that only 8% to 12% of type 1 GD patients shows PD symptoms at age 80 years (Blauwendraat et al., 2023).
Line 48: It is not specified which is the most common type of GD
Corrected.
In table 1: is it necessary to specify how the dispersion measures of the mean (SD, SE, range) are reported?
The sentence “Values represent mean ± S.E.M.” was added.
Table 3 should also report the duration of treatment for both the ERT and SRT groups: an explanation of the results obtained could be that while the ERT group had been on treatment for a long time and therefore reached a plateau for which it is more difficult to notice changes, the SRT group started therapy more recently, so the 30 months of observation looked at the initial effect of the drug before reaching plateau.
Thank you for your comment. We fully agree with your explanation that the bias caused by more recent introduction of SRT might lead to better results and long treated patients on ERT. Therefore we were very careful in interpretation of possible advantage of SRT over ERT in the text in following way (s. limitations paragraph in discussion): “We proved non-inferiority of SRT; on the contrary, our results indicate that ERT and SRT could offer potential of individual drug effectiveness in treating particular symptoms, either as a combination or in a switch manner.“
We also attenuated the sentence: „Association of Lyso-Gb1 with evaluated markers was markedly stronger in the SRT cohort“ by deleting the work markedly. The information that “All patients were on long-term ERT therapy with mean time of years in treatment of 15.6 (range 2.0 to 25.0)” is shown in the text.
In Table 1, did the p-value refer to the differences between ERT and SRT? You should divide this part from the rest of the table. Furthermore, this information is quite imprecise because it seems that SRT was more effective, while this depends on the baseline values of the parameters evaluated, on the age of the patient being treated.
Thank you. After consideration of your important comment we omitted the statistical comparison of the baseline values as imprecise and deleted it from the table. The parts of the table are graphically separated by and empty line.
In Table 2, instead of the total dose of ERT per patient, it would be appropriate to report the dose per kg of body weight administered every 2 weeks. The same should be done in the article.
Thank you for this important comment. As we used the standard recommended therapeutic dosage for all patients, we added the following sentence in 2.1. Study population description.
The therapy was given in the standard dosage as recommended by drug prescribing information based on the weight of patient (ERT) or CYP2D6 activity (SRT).
The following sentence was deleted as imprecise:
The ERT patients were receiving a median dose of 2000 U/infusion (range 400-2800 U/infusion every 2 weeks) albeit with different ERT preparations. The SRT patient were dosed with 84 mg of eliglustat (Cerdelga®) or with 200 mg of miglustat (Zavesca®) twice daily.
We also deleted the redundant and imprecise last column with the whole dosage in the Table 2.
Figures 1 and 2 are not drawn correctly and should be revised to make them more readable.
We have newly uploaded the original detailed images and asked the editor for graphic editing.
Your bone metabolism data was mentioned in the discussion (line 351), in this case this data should be shown.
The remark was based on our shown-above TRAP data.
Even if the number is low, perhaps the patient who switched from ERT to SRT should be eliminated from the analysis because it is not clear in which group he was analysed.
The patient was analysed in the SRT group, using just data from his SRT treatment period.

Round 2
Reviewer 2 Report
Dear Authors,
although this version of the article has improved considerably, there are still some points (already underlined in the previous revision) that need to be clarified:
Table 3 must report the duration of treatment for both the ERT and SRT groups. Apart from the explanation of why there are these differences between the two groups, I believe it is necessary to report this information to allow the reader to understand the different clinical history of the patients.
In Table 2 it would be appropriate to report the dose per kg of body weight administered every 2 weeks: because the dosages of ERT can be personalized according to the clinical response, while those of SRT are stable over time. The same should be done in the article.
Minor editing of English language required
Author Response
Dear reviewer,
Thank you for your comment and insisting on your previous comments. We do really appreciate your approach, as now we feel that the idea of the article is now presented more clearly.
We have added medication dosage to the Table 2, length of therapy for both ERT and SRT cohorts in Table 3. We have also added information about the duration of therapy to the Result section:
Prior to entering the study, patients on ERT therapy had been treated for 15.6 years in the average (range 2.0 to 25.0) and patients on SRT 2.8 years (range 0.0 to 11 years), respectively.
To the discussion:
Our results revealed that in patients with long-term therapy (the mean of years in treatment in the ERT cohort was 15.6 years), a decline in CHITO levels is still evident. However, it reached statistical significance emphasized with the skewer slope of the regression curve only within the SRT cohort (the mean of the previous treatment was 2.8 years).
And to the limitations of the study:
The present study has several limitations, and the results should be interpreted with caution. First, our study sample is of a very limited size. Moreover, the ERT to SRT patient ratio is highly inequal (6 to 1) and differs in the duration of treatment before entering the study. One of the further limitations of the study was the inclusion of a higher ratio of females to males (24F/11M), reflecting our center’s patient population. Another limitation of the study was some heterogeneity of patients in treatment dosage and with wide variation in GBA1 mutation.
The English of the manuscript was corrected by a native-speaker (tracked changes).
